# Fabrication and Characterization of Gel-Forming Cr$_2$O$_3$ Abrasive Tools for Sapphire Substrate Polishing

**Liang Zhao [1,2], Kaiping Feng [1,2,3,*], Binghai Lyu [2,*], Tianchen Zhao [1,3] and Zhaozhong Zhou [1,3]**

1   College of Mechanical Engineering, Quzhou University, Quzhou 324000, China
2   Ultra-Precision Machining Centre, Zhejiang University of Technology, Hangzhou 310014, China
3   Key Laboratory of Air-Driven Equipment Technology of Zhejiang Province, Quzhou University, Quzhou 324000, China
*   Correspondence: 36065@qzc.edu.cn (K.F.); icewater7812@126.com (B.L.); Tel.: +86-182-6896-2302 (K.F.)

**Featured Application: This research explores the optimal machining parameter combination for a gel abrasive tool and proves that a gel abrasive tool has a better processing effect than a hot-pressing tool on the semi-finishing of sapphire substrate.**

**Abstract:** This paper proposes a gel-formed abrasive tool to address the problem of abrasive agglomeration in a traditional hot-pressing abrasive tool. The effect of Polyimide resin content on the mechanical properties of the gel abrasive tools were tested, and a comparison of the mechanical properties of the gel abrasive tool and the hot-pressing tool was conducted. An orthogonal experiment was conducted to explore the best combination of machining parameters. A polishing experiment of sapphire was conducted to compare the processing effect of the gel abrasive tool and hot-pressing tool. The results from testing the mechanical properties showed that the tensile, flexural, and impact strength of the gel abrasive tool was better than that of the hot-pressing abrasive tool. The results of the orthogonal experiment showed that the best process parameters of the gel abrasive tool were a spindle speed of 900 rpm, a feed rate of 8 μm/min, and a grinding depth of 16 μm. The polishing experiment showed that the gel abrasive tool had a better processing effect on sapphire. The sapphire surface processed by the gel abrasive tool had no deep scratches, and an ultrasmooth surface could be obtained after chemical mechanical polishing (CMP).

**Keywords:** polyacrylamide gel; polyimide resin; Cr$_2$O$_3$ abrasives; sapphire substrate; surface roughness

## 1. Introduction

Sapphire is widely used in optical instrument manufacturing and electronic technology because of its excellent mechanical, optical, and thermal properties. For example, sapphire can be used as an epitaxial material for growing LED lamps and epitaxial wafers on gallium nitride substrates. Sapphire is a hard and brittle material that is difficult to process. To improve the processing effect, the importance of the machining tool's wear performance should be considered. For example, Macerol et al. [1] investigated the wear behavior of bonded abrasive tools using a lapping-based test method. Li et al. [2] explored the wear performance of alumina abrasive wheels used to process a superalloy. Sani et al. [3] explored the effect of the pressure gradient and abrasive tool wear when polishing ceramic tiles. The preparation method also plays a significant role. For example, Mayank et al. [4] developed a flexible abrasive tool, while Wang et al. [5] compared the polishing effect of single alumina abrasive grains and alumina/metatitanate abrasives with a core–shell structure on sapphire substrates, and they found that the abrasives with a core–shell structure could not only increase the MRR but also yield a better surface quality. Liu et al. [6] fabricated a hydrophobic fixed abrasive pad using a layer-by-layer method. Feng et al. [7] prepared fixed agglomerated diamond pads and explored the friction and wear characteristics of agglomerated diamond abrasives. To achieve a damage-free surface, chemical mechanical

polishing (CMP) is an effective method. Silica oxide ($SiO_2$) abrasive is widely used in CMP due to its low hardness, as well as good stability and dispersion. Yan et al. [8] explored the effect of different Fenton reagents on the stability of a silica sol polishing solution, as well as their influence on the polishing effect of SiC wafers; it was found that, when $H_2O_2$ was added to the silica sol polishing solution, the stability of the polishing solution improved, and the surface of the processed SiC wafer was smooth with a surface roughness of 0.4642 nm. Liu [9] predicted the material removal rate (MRR) of CMP using a fusion network. Xiong et al. [10] used silica abrasives with different particle sizes to polish sapphire, and they found that the MRR increased with the increase in particle size of silica; however, silica with an excessive particle size led to a reduction in the surface quality of sapphire after processing. The quality of sapphire obtained using the $SiO_2$ abrasive with a particle size of 80 nm was best. Sun et al. [11] compared the processing effect on sapphire wafers of a single silica abrasive with a consistent particle size or mixed particle size. Xu et al. [12] proposed a method for polishing sapphire using SoFe$^{III}$ as a catalyst, and they found that, when the polishing solution contained SoFe$^{III}$ at $-80$ °C, the MRR reached 7.21 μm/h. Although the abovementioned methods improved the CMP method, the processing efficiency remains a big problem.

Fixed abrasive tools have the advantages of a high material removal rate and high surface accuracy of the workpiece, and they are widely used in the semi-finishing process. However, the preparation process of traditional fixed abrasive tools involves dry mixing, resulting in the formed tool having the problems of uneven relative density, large differences in microhardness, and serious abrasive agglomeration. The above problems may lead to a short service life of the abrasive tool and a poor surface quality of the workpiece. Zhong et al. [13] studied the effect of adding different volume fractions of sodium chloride crystals to phenolic resin on the polishing performance of granite, and they found that, when the particle size of sodium chloride was 250–500 nm and the volume fraction was 20%, the effect was best after processing. Chen et al. [14] proposed a method of adding soft aluminum abrasive grains to a fixed diamond abrasive tool for processing sapphire, and they finally obtained a sapphire wafer with high surface quality. Wu et al. [15] prepared a polysaccharide binder abrasive tool to process sapphire, and they studied the effect of the sand-to-bond ratio on the performance of the abrasive tool. After 10 min of processing, the surface roughness $R_q$ value of sapphire decreased by 84.25%. Zhao et al. [16] prepared a polishing slurry with manganese oxide particles for polishing SiC substrates. Zhang [17] proposed green CMP by combining silica nanoparticles of 50 nm, triethanolamine (TEA), sodium metasilicate nonahydrate, and deionized water as the polishing slurry. Feng et al. [18–21] used PVA/PF composite gel as the bonding agent to prepare a diamond abrasive tool, which was combined with CMP to process a silicon carbide wafer. Wang et al. [22] prepared a diamond/epoxy abrasive tool, and the wear resistance of the abrasive tool was improved by adding graphene oxide. Lu [23] used $SiO_2$-coated diamond abrasives to polish a SiC wafer, and the results showed that the modified abrasives had a higher material remove rate and could yield a better surface quality of the SiC wafer. Huang [24] developed an abrasive tool using the sol–gel process with ceramic corundum as an abrasive for machining hard materials.

The semi-finishing process is important for hard and brittle materials, as it can avoid the aggregation of processing stresses while improving the CMP efficiency. To further promote the surface quality of the workpiece, the machining tool plays a significant role in processing. This paper proposes a gel-formed $Cr_2O_3$ abrasive tool and tests its performance and mechanical properties in comparison with a traditional hot-pressing tool. Since the machining conditions have a great effect on the surface quality of the workpiece, orthogonal experiments were conducted to explore the best combination of machining parameters. The processing effect of hot-pressing abrasive tools and gel abrasive tools on sapphire was compared. The damage of the gel abrasive tools at different processing stages is revealed, and the corresponding trimming method is given.

## 2. Abrasive Tool Preparation

In this paper, the sol–gel method was used to prepare the abrasive tool. Polyacrylamide gel has a good three-dimensional network structure, and its gelation process is a typical free-radical polymerization reaction, which is the result of one branched structure diffusing and apportioning the entire space in the gel system [25]. The mixed solution of bulk acrylamide (AM) and crosslinking agent methylene bisacrylamide (MBAM) has low viscosity, which is conducive to the dispersion of inorganic particles in it. Thermosetting polyimide resin has the advantages of good mechanical properties and high temperature resistance. Wu et al. [26–28] found through experiments that, when using $Cr_2O_3$ as an abrasive to process sapphire, the solid-phase chemical reaction of the two was strong due to the similar crystal structures of hexagonal crystals of chromium oxide and aluminum oxide; therefore, $Cr_2O_3$ was selected as the experimental abrasive. The slurry needs to be ball-milled and dispersed before initiation. The gel wraps abrasives and fillers in the initiation process to form a green body and is sintered at 350 °C to form an abrasive tool.

The composition of the abrasive tool is shown in Table 1. The wetting agent can improve the wetting ability of the glue to the inorganic particles. Polyethylene glycol (PEG) has good wetting, moisturizing, and dispersing properties, while dibutyl ester has good plasticity, which can enhance the gelling ability of the solution. The dispersant can chemically modify the surface of $Cr_2O_3$ abrasives, which avoids the abrasive agglomeration. The preparation process is shown in Figure 1.

**Table 1.** Contents of $Cr_2O_3$ abrasive tool (before sintering).

| Component | Solid Content (wt.%) |
| --- | --- |
| PAM + PI | 15 |
| PEG400 | 1 |
| Wetting agent | 0.5 |
| Dispersant | 1.5 |
| Defoamer | 1 |
| Dibutyl ester | 0.5 |
| $Cr_2O_3$ Powder | 82.5 |

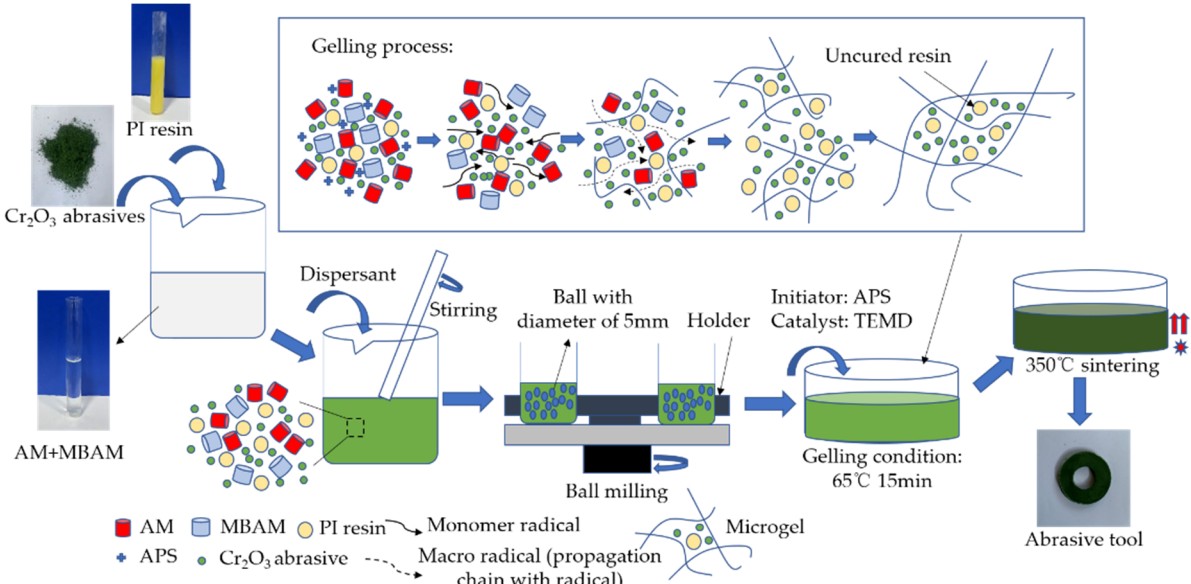

**Figure 1.** The preparation process of gel abrasive tools.

## 3. Experimental Setup and Equipment

This experiment used sapphire substrates as the processing object. Before the polishing experiment, the sapphire was pre-machined using a diamond abrasive tool, and its surface

roughness reached 54 nm. The precision polishing experiment was carried out on a grinding machine with a power monitoring system (Beckhoff system); the machining parameters are shown in Table 2. In order to explore the optimal processing parameters of the gel abrasive tool, an orthogonal experiment was designed. To compare the processing effects of the gel abrasive tool and hot-pressing abrasive tool, a comparative experiment was carried out. The orthogonal experiment parameters are shown in Table 3. The three factors were the spindle speed ($N$), the grinding depth ($D$), and the feeding rate ($V_f$). The spindle speeds ($n_1$–$n_3$) were 300 rpm, 600 rpm, and 900 rpm, the grinding depths ($d_1$–$d_3$) were 8 μm, 16 μm, and 24 μm, and the feed rates ($f_1$–$f_3$) were 8 μm/min, 12 μm/min, and 0.016 μm/min. After the experiment, the degree of influence of each process parameter on the machining result was determined by range analysis, and the model of surface roughness was predicted and calculated using the method of linear regression.

**Table 2.** Machining parameters.

| Condition | Value |
| --- | --- |
| Coolant | Water |
| Workpiece diameter | 100 mm |
| Rotating speed (workpiece) | 150 rpm |
| Spindle speed (rpm) | 700, 750, 800, 850, 900, 950, or 1000 |
| Feeding rate (μm/min) | 2, 4, 6, 8, 10, 12, or 14 |
| Grinding depth (μm) | 8, 10, 12, 14, 16, 18, or 20 |
| Processing time (min) | 30 |

**Table 3.** Orthogonal experiment parameters.

| Group | $N$ (rpm) | $D$ (μm) | $V_f$ (μm/min) |
| --- | --- | --- | --- |
| 1 | 300 | 8 | 8 |
| 2 | 300 | 16 | 16 |
| 3 | 300 | 24 | 12 |
| 4 | 600 | 8 | 16 |
| 5 | 600 | 16 | 12 |
| 6 | 600 | 24 | 8 |
| 7 | 900 | 8 | 12 |
| 8 | 900 | 16 | 8 |
| 9 | 900 | 24 | 16 |

The MicroXAM 1200 white-light interferometer from Taiwan China was used to test the micromorphology and surface roughness of the workpiece after polishing. Ten points were taken along two vertical diameter directions, and the flatness was measured using HSINTEK AK100F3 laser interferometry from Taiwan China. The measurement method is shown in Figure 2. The KEYENCE CL-3000 laser displacement sensor from Japan (accuracy 0.25 μm) was used to achieve precise measurement of the workpiece grinding depth; the Fluke Thermal imaging camera from U.S. was used to observe the thermogram.

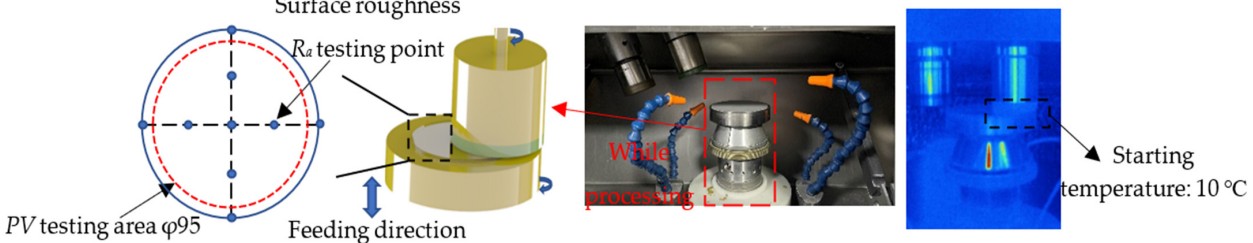

**Figure 2.** Measurement method and machining diagram.

## 4. Results and Discussion

### 4.1. Comparison of Mechanical Properties

Figure 3 shows the SEM microtopography of the gel abrasive tool and the hot-pressing abrasive tool. The agglomerated abrasives can be observed in the micrograph of the hot-pressing abrasive tool. On the one hand, the preparation method of the hot-pressing abrasive tool involves dry mixing, whereby the fluidity of the mixed materials is much worse than that of the wet mixing method. Furthermore, there are density differences between different materials, which renders the mixing process more difficult. On the other hand, the abrasives have a high surface energy and large contact surface due to the small particle size, whereby agglomeration is prone to happen due to van der Waals forces between the particles. It can be observed from Figure 3a that abrasives were uniformly dispersed in the gel abrasive tool because the premixed solution of the gel abrasive tool had low viscosity. Moreover, under the action of chemical reagents such as the dispersant and wetting agent, the abrasive had a better dispersion effect. Pore structure is an important feature of abrasive tools; Archimedes' principle was used to test the density and porosity of samples. The porosity of the abrasive tool can be obtained by calculating the theoretical density and weighting dry weight, floating water weight, and water weight of the sample. The existence of pores in the abrasive tool enables the abrasive tool to have good debris-holding and heat-dissipation capabilities. It can be seen that there were many uniformly distributed pores on the surface of the gel abrasive tool, which were caused by the loss of water during the drying process.

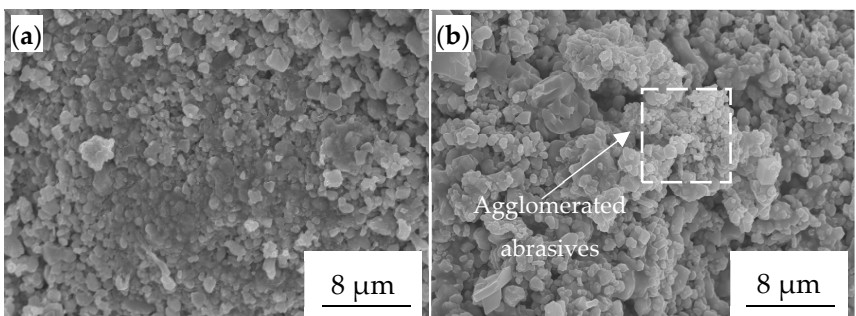

**Figure 3.** Surface SEM microtopography: (**a**) gel abrasive tool; (**b**) hot-pressing abrasive tool.

The DUH–211/211S Dynamic Microhardness Tester was used to establish force–displacement curves; the maximum test force was 500 mN and the loading speed was 7.0067 mN/s. Five points were chosen to test the microhardness, and the elastic modulus was obtained after the test. Figure 4 shows a comparison of the microhardness and elastic modulus of the gel abrasive tool and the hot-pressing abrasive tool; the results show that the microhardness of the hot-pressing tool fluctuated greatly due to the appearance of loosely distributed agglomerated particles. The flexural strength was tested using the three-point bending method; the span distance was about 60% of the sample's length, and the speed was 1 mm/min. The tensile strength was tested using the uniaxial stretching method; the stretching speed was 1 mm/min. The impact strength was tested using the pendulum impact method; the impact rate was 3.5 m/s, the pendulum potential energy was 5 J, the sample size was 10 mm × 10 mm × 50 mm, and the surface roughness Ra of the sample was about 1 μm. Figure 5a shows the effect of PI resin content on the mechanical properties of the gel abrasive tool, where it can be seen that the mechanical properties showed an upward trend with the increase in mass fraction of the PI resin; however, when the mass fraction of PI resin was too high, the viscosity of the slurry was significantly affected, which worsened the uniformity of the abrasive tool. As shown in Figure 5b, the flexural and tensile strength of the gel abrasive tool was 25% and 23% higher than that of the hot-pressing abrasive tool, respectively, while the impact strength was 6% higher. The curing temperature of PI resin was much higher than the gelling temperature, which

enabled the gel to wrap the uncured resin and abrasive particles in the uniformly dispersed microgel during the gelling process. During the sintering process, the gel component in the abrasive tool began gradually decomposing, and the polyimide resin gradually became the main body of the bonding agent. Accordingly, the gel abrasive tool showed better mechanical properties, whereas, in the dry-mixed hot-pressing abrasive tool, the relative density of each part of the tool was inconsistent, which likely negatively impacted its mechanical properties.

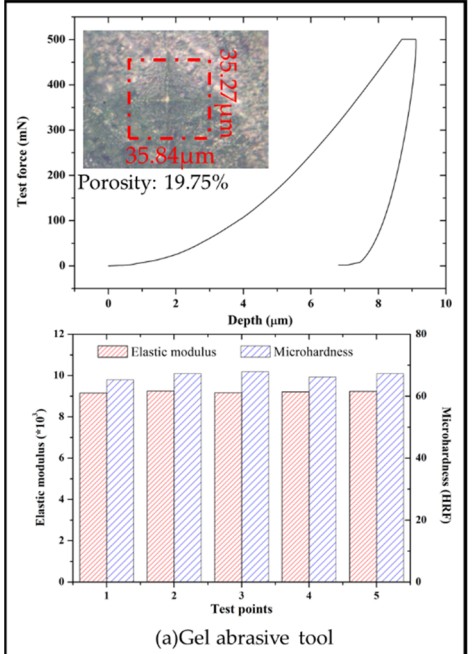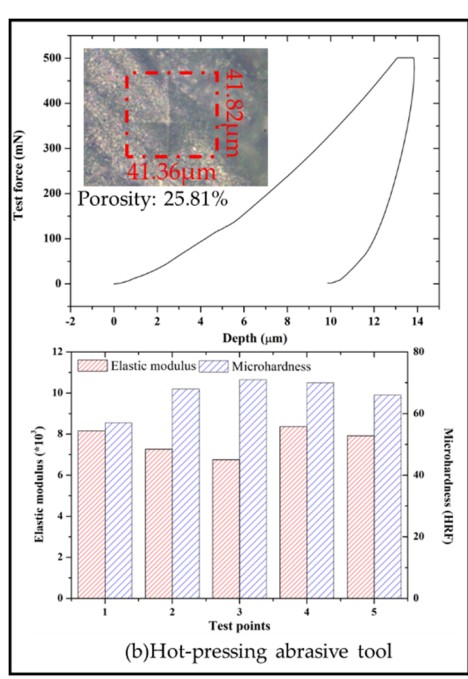

**Figure 4.** Force–displacement curves: (**a**) gel abrasive tool; (**b**) hot-pressing abrasive tool.

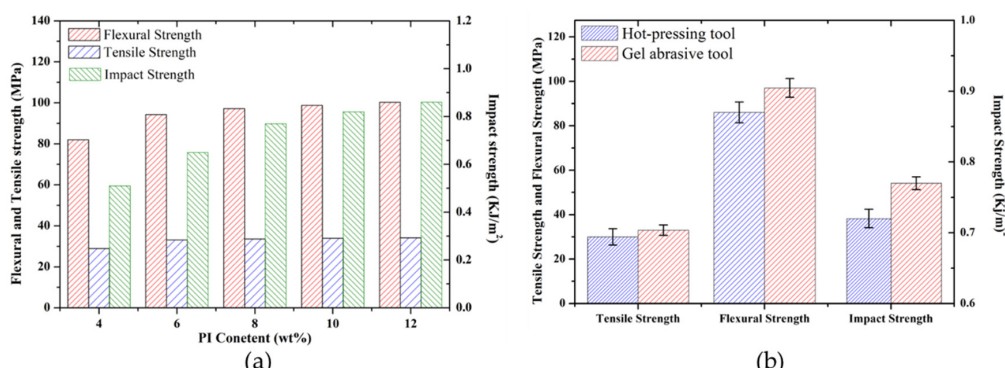

**Figure 5.** (**a**) The influence of PI resin content on the mechanical properties. (**b**) A comparison of the mechanical properties of the gel abrasive tool and hot-pressing tool.

### 4.2. Analysis of Orthogonal Experiment Results

Results of the orthogonal experiment are shown in Table 4, and the analysis of the extreme deviation of the surface roughness of the workpiece after machining is shown in Table 5. Here, $K_1$ denotes the sum of the results ($R_a$) when the machining parameters were $n_1$, $d_1$, and $f_1$, and $K_{a1}$ denotes the average results when the machining conditions were $n_1$, $d_1$, and $f_1$; the values of $K_2$, $K_3$, $K_{a2}$, and $K_{a3}$ could be calculated in the same way, thereby obtaining the max–min value. The results show that the degree of influence of various factors on the surface roughness of the workpiece in descending order was as follows: feeding rate > grinding depth > spindle speed. The best combination of process parameters was a spindle speed of 900 rpm, feeding rate of 8 μm/min, and grinding depth of 16 μm.

**Table 4.** Orthogonal experiment results.

| Group | N | D | $V_f$ | $R_a$ (nm) |
|---|---|---|---|---|
| 1 | n1 | d1 | f1 | 2.215 |
| 2 | n1 | d2 | f3 | 3.076 |
| 3 | n1 | d3 | f2 | 2.886 |
| 4 | n2 | d1 | f3 | 2.563 |
| 5 | n2 | d2 | f2 | 2.330 |
| 6 | n2 | d3 | f1 | 2.556 |
| 7 | n3 | d1 | f2 | 2.330 |
| 8 | n3 | d2 | f1 | 1.880 |
| 9 | n3 | d3 | f3 | 3.226 |

**Table 5.** Analysis of extreme deviation.

| Results | N | D | $V_f$ |
|---|---|---|---|
| $K_1$ | 8.177 | 7.108 | 6.651 |
| $K_2$ | 7.449 | 7.286 | 7.546 |
| $K_3$ | 7.436 | 8.668 | 8.865 |
| $K_{a1}$ | 2.725 | 2.359 | 2.217 |
| $K_{a2}$ | 2.483 | 2.428 | 2.515 |
| $K_{a3}$ | 2.478 | 2.889 | 2.955 |
| Max–Min | 0.247 | 0.520 | 0.738 |

The traditional empirical formula of surface roughness is usually expressed as an exponential function, as shown in Equation (1); hence, a prediction model containing three parameters can be established [29].

$$R_a = CN^{b1} D^{b2} V_f^{b3} \tag{1}$$

where $C$ is the proportional coefficient, $N$, $D$, and $V_f$ refer to the spindle speed, grinding depth, and feeding rate, respectively, and $b_1$, $b_2$, and $b_3$ refer to the indices of each variable, respectively. The logarithm of both sides can be obtained to yield Equation (2).

$$ln\,(R_a) = ln\,(C) + b_1 ln\,(N) + b_2 ln\,(D) + b_3 ln\,(V_f) \tag{2}$$

where it can be defined that $y = ln\,(R_a)$, $b_0 = ln\,(C)$, $x_1 = ln\,(N)$, $x_2 = ln\,(D)$, and $x_3 = ln\,(V_f)$, thereby yielding Equation (3).

$$y = b_0 + b_1 x_1 + b_2 x_2 + b_3 x_3 \tag{3}$$

The above linear equation contains a total of three independent variables, and the test results can be expressed in terms of y. The independent machining parameters of the i-th group are $x_{i1}$, $x_{i2}$, and $x_{i3}$ (where $x_{11}$ denotes n1, $x_{12}$ denotes d1, and $x_{13}$ denotes f1), and the test results are $y_i$. The following multiple linear regression equation can be established:

$$\begin{cases} y_1 = \beta_0 + \beta_1 x_{11} + \beta_2 x_{12} + \beta_3 x_{13} \\ y_2 = \beta_0 + \beta_1 x_{21} + \beta_2 x_{22} + \beta_3 x_{23} \\ \qquad\qquad \cdots \\ y_9 = \beta_0 + \beta_1 x_{91} + \beta_2 x_{92} + \beta_3 x_{93} \end{cases} \tag{4}$$

Equation (4) can be expressed in matrix form as

$$Y = \beta X$$

The orthogonal experiment results were introduced into the formula, and the regress function in the MATLAB was used to calculate the surface roughness residual map and the indices $C$, $b_1$, $b_2$, and $b_3$. The results are shown in Figure 6. The points in the figure

represent the residuals of the data, and the line segments represent the confidence intervals. It can be seen that each point was relatively close to the zero line, and the confidence intervals all included the zero-point line, indicating that the model could predict the surface roughness of sapphire. The indices calculated using the regress function were as follows: $C = 1.0857$, $b_1 = -0.0890$, $b_2 = 0.1580$, and $b_3 = 0.4052$; therefore, the prediction model could be expressed as

$$R_a = 1.0857 N^{-0.0890} D^{0.1580} V_f^{0.4052}$$

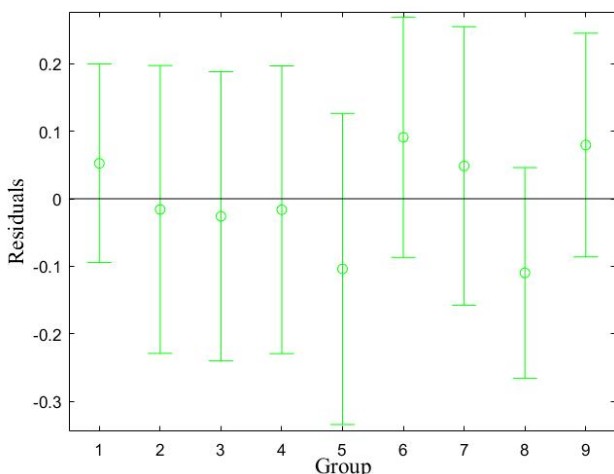

**Figure 6.** Residual map of the surface roughness.

Figure 7 compares the data points in the curve obtained according to the prediction model and the actual measured data points. It can be seen that the prediction model was basically consistent with the actual measured values.

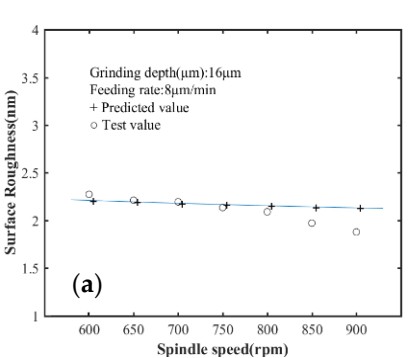 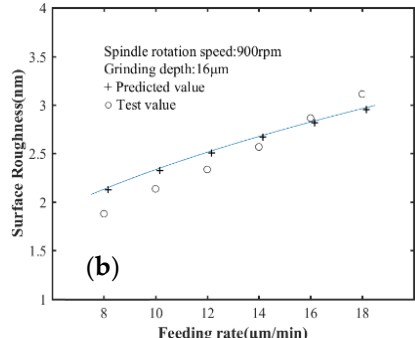 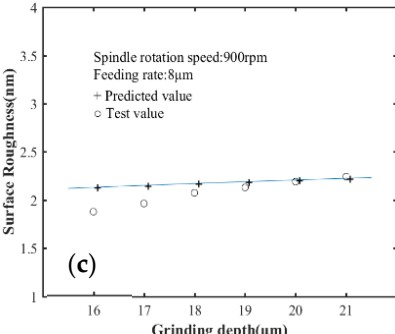

**Figure 7.** Influence of different machining parameter on the surface roughness Ra: (**a**) spindle speed; (**b**) feeding rate; (**c**) grinding depth.

### 4.3. Influence of Machining Parameters on Surface Roughness

The material removal rate was calculated as a function of the thickness change of the sapphire. Figure 8a shows the relationship between spindle speed and surface roughness, while Figure 8b shows the influence of spindle speed on the machining temperature. Figure 9 shows the surface topography of the sapphire under different spindle speeds. It can be seen that the roughness of sapphire first decreased and then increased with the change in spindle speed. On the one hand, this was due to the fact that, as the spindle speed increased, the linear speed of the grinding abrasive increased, the quantity of abrasive involved in the grinding increased, and the grinding force and depth requirement of a single abrasive decreased; accordingly, the workpiece's plastic deformation also decreased, resulting in a decrease in surface roughness [30]. On the other hand, with the increase in spindle speed, more coolant entered the same area simultaneously, which could reduce

the temperature of the grinding area and prevent damage and glazing of the abrasive tool g. The number of abrasives involved in grinding became excessive as the spindle speed increased, making it difficult to remove the resulting debris from the processing region on time, causing abrasive debris to clog the pores and overheat the abrasive tool, and decreasing the mechanical effect of the abrasive tool. Adhesive wear reduced the surface quality of the sapphire substrate. Figure 8a shows the relationship between the rotational speed and the material removal rate. It can be seen that the material removal rate increased first and then decreased with the increase in spindle speed. The number of abrasives involved in grinding increased with spindle speed, as did the material removal rate. Nevertheless, when the spindle speed was too high, the number of abrasives involved in grinding decreased, while the abrasive tool tended to glaze, which reduced its ability to self-sharpen; accordingly, the rate of material removal decreased.

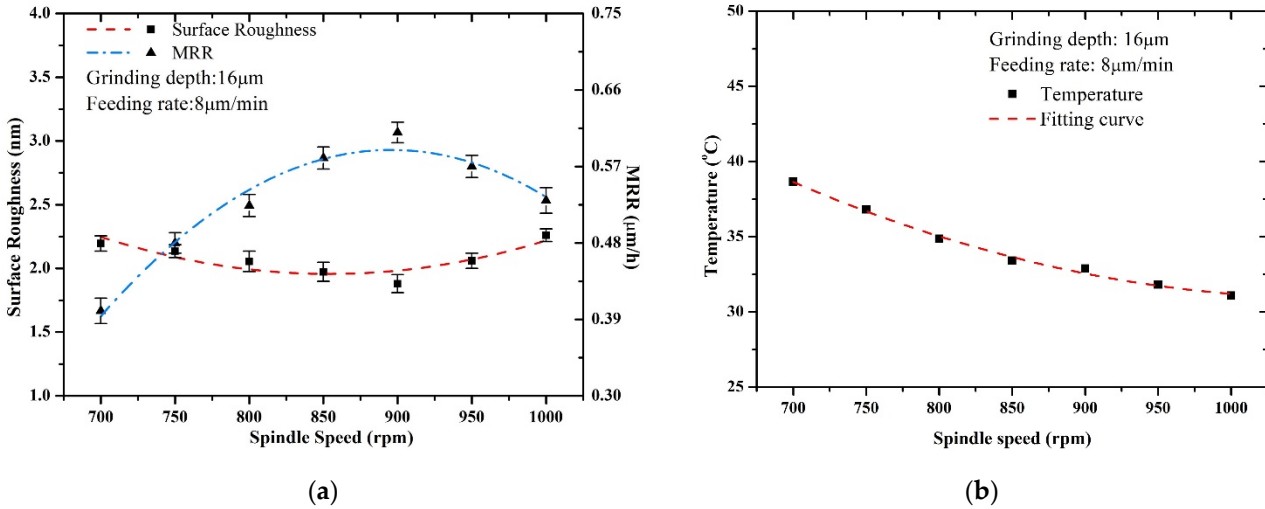

**Figure 8.** Influence of spindle speed on the (**a**) surface roughness, MRR, and (**b**) machining temperature (average temperature in the contact area after machining for 5 min).

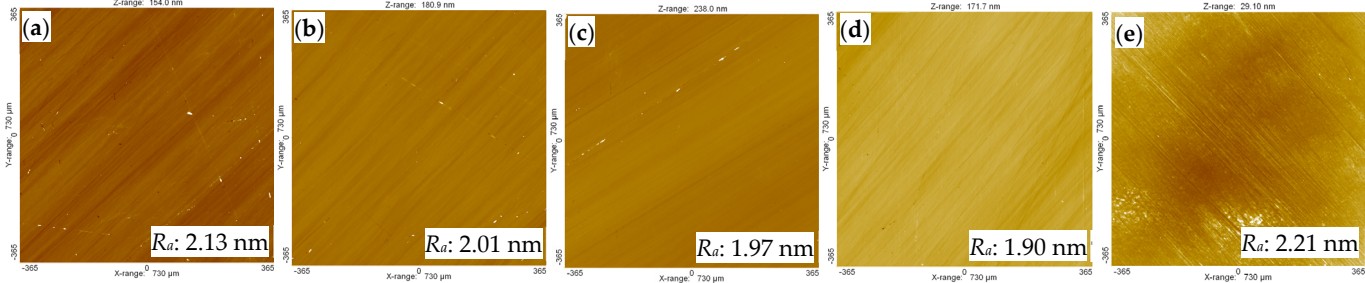

**Figure 9.** Surface topography of sapphire under different spindle speeds: (**a**) 750 rpm; (**b**) 800 rpm; (**c**) 850 rpm; (**d**) 900 rpm; (**e**) 950 rpm.

Figure 10a shows the relationship between feeding rate and surface roughness, while Figure 10b shows the influence of feeding rate on the machining temperature. Figure 11 shows the surface topography of sapphire under different feeding rates. It can be seen that, with the increase in feeding rate, the roughness of sapphire initially decreased and then increased. When the feeding rate was 4 μm/min, the surface roughness was highest. On the one hand, the tangential grinding force of the tool was small, and the grinding traces could hardly be observed from Figure 11a, indicating a poor mechanical effect of the abrasive tool. On the other hand, it can be seen from Figure 10b that the temperature of the contact area was low; since the solid-phase chemical reaction is closely related to the temperature [26], when the feeding rate was too low, the temperature struggled to increase with the effect of coolant. When the feeding rate increased, the temperature

also increased due to the increase in pressure between the sapphire and the abrasive tool, thereby promoting solid-phase chemical reactions; hence, the surface roughness presented a descending trend. When the feeding rate exceeded 8 μm/min, the surface roughness worsened, partly due to the accelerated appearance of a glazing film with the increase in feeding rate, which blocked the abrasives and weakened the reaction, and partly due to the accelerated accumulation of frictional heat, which weakened the holding ability of the binder. This led to the abrasives falling off prematurely, resulting in a significant fluctuation in the friction coefficient between the workpiece and the abrasive tool, which harmed the surface roughness of the sapphire. Figure 10a shows the relationship between feeding rate and material removal rate. It can be seen that, with the increase in feeding rate, the material removal rate showed an upward trend. When the feeding rate was low, the chemical and mechanical effects were both weak. With the increase in feeding rate, the temperature of the interaction area gradually increased, thereby promoting the solid-phase chemical reaction; hence, the material removal rate increased.

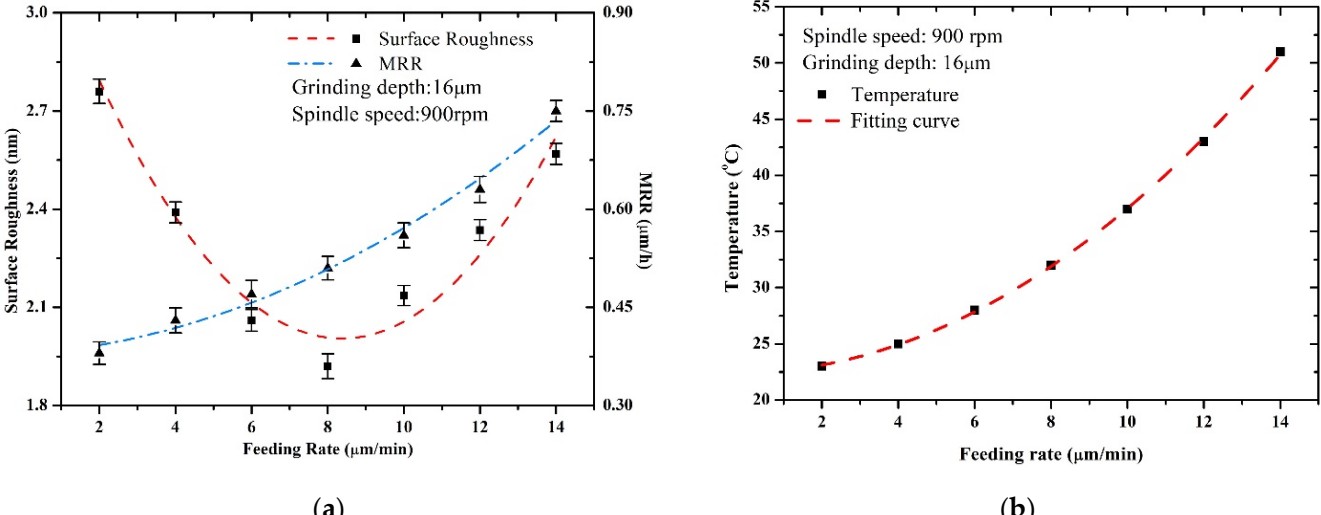

(**a**)          (**b**)

**Figure 10.** Influence of feeding rate on the (**a**) surface roughness, MRR, and (**b**) machining temperature (average temperature in the contact area after machining for 5 min).

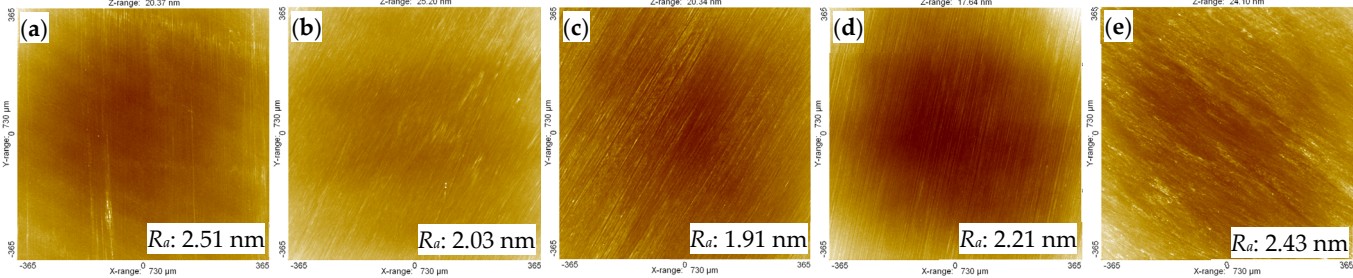

**Figure 11.** Surface topography of sapphire under different feeding rates: (**a**) 4 μm/min; (**b**) 6 μm/min; (**c**) 8 μm/min; (**d**) 10 μm/min; (**e**) 12 μm/min.

Figure 12a shows the relationship between grinding depth and surface roughness, while Figure 12b shows the influence of grinding depth on the machining temperature. Figure 13 shows the surface topography of sapphire under different grinding depths. In this paper, the grinding depth referred to the feeding distance of the tool when it first touched the sapphire; due to abrasive tool wear, the grinding depth was slightly less than the theoretical value. It can be seen from Figure 14 that the surface roughness of sapphire decreased with the increase in grinding depth and then increased. When the grinding depth was low, the pressure between the abrasive tool and sapphire was slight, and it can be seen from Figure 13a that only some shallow grinding traces were left on the sapphire. Furthermore, from Figure 12b, it can be seen that the machining temperature was low in the contact area, which was not conducive to the solid-phase chemical reaction. Upon increasing the grinding depth, the contact area of the grinding zone increased, the number of abrasive particles participating in the grinding per unit area increased, and the heat accumulation caused the temperature of the grinding zone to rise rapidly [19]. Within a certain temperature range, an increase in temperature was beneficial to increase the solid-phase chemical reaction rate, which increased the plastic removal ratio of the abrasive tool and improved the surface quality of the sapphire. When the grinding depth was further increased, the volume of material removed by the abrasive tool increased, while the friction area increased gradually, the grinding force perpendicular to the surface of the workpiece increased, the grinding force required by a single abrasive particle increased, and the mechanical effect of the abrasive tool began to dominate, which increased the surface roughness. Figure 12a shows the relationship between the grinding depth and the material removal rate. As the grinding depth increased, the contact area between the abrasive tool and the workpiece increased, the rate of frictional heat accumulation increased, and the material removal rate of the abrasive tool increased, as the increase in temperature could promote the solid-phase chemical reaction. When the grinding depth was further increased, the rapid accumulation of frictional heat led to rapid glazing of the abrasive tool, causing a reduction in the chemical and mechanical effects, and the material removal rate accordingly began decreasing.

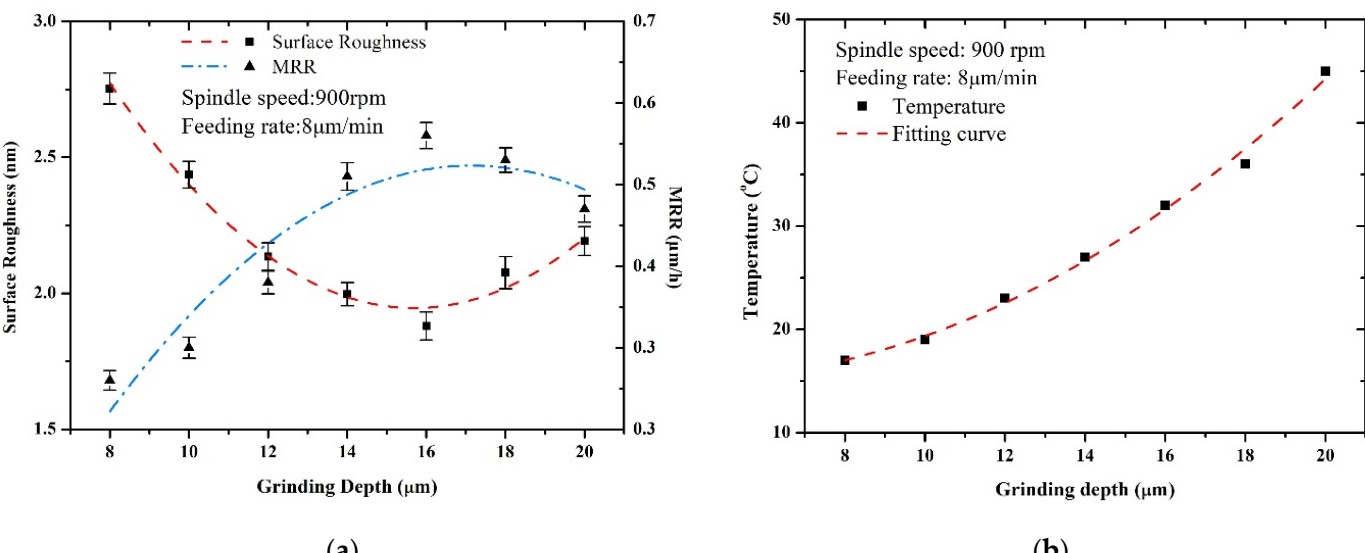

(a)    (b)

**Figure 12.** Influence of grinding depth on the (**a**) surface roughness, MRR, and (**b**) machining temperature (average temperature in the contact area after machining 5 min).

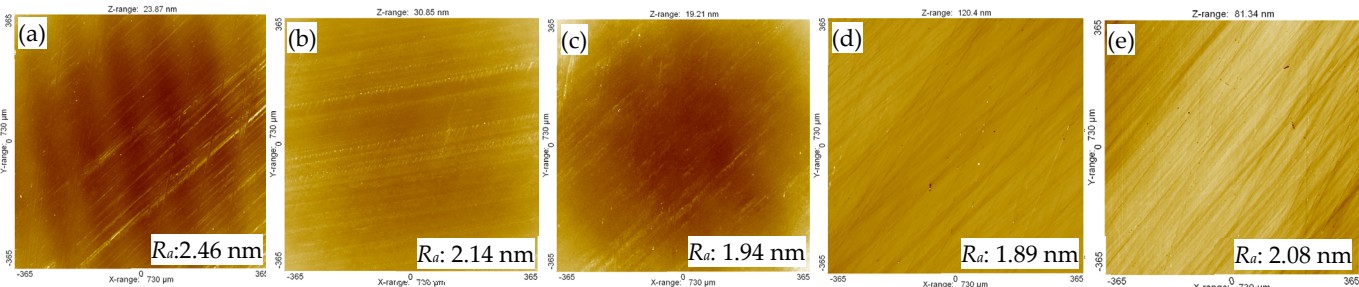

**Figure 13.** Surface topography of sapphire under different grinding depths: (**a**) 10 μm; (**b**) 12 μm; (**c**) 14 μm; (**d**) 16 μm; (**e**) 18 μm.

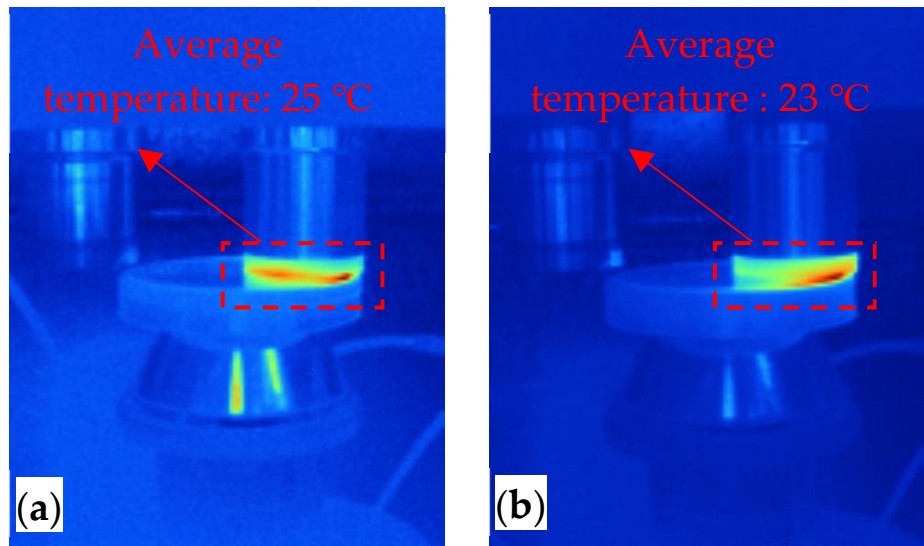

**Figure 14.** Thermogram of different tools (after machining 5 min): (**a**) gel abrasive tool; (**b**) hot-pressing abrasive tool.

*4.4. Comparative Processing Experiments*

The hot-pressing tool and gel abrasive tool were used to process the sapphire. Some parameters in the above experiments were selected to carry out comparative experiments of the two abrasive tools. The feed rate was 8 μm/min or 10 μm/min, and the grinding depth was 12 μm or 16 μm. The processing results are shown in Table 6. The surface roughness of the sapphire substrate processed by the gel abrasive tool in each group was better than that processed by the hot-pressing abrasive tool. The tool wear rate and the MRR results are shown in Table 7; the tool wear rate was measured as a function of the thickness change of the abrasive tool. The results show that the wear rate of the hot-pressing abrasive tool was larger; the main reason is that the abrasive holding ability of the gel abrasive tool was much stronger than that of the hot-pressing abrasive tool, and the material removal rate of the hot-pressing abrasive tool was slightly larger than that of the gel abrasive tool due to the high concentration of abrasives, which promoted the solid-phase chemical reaction. Figure 14 shows the thermogram of different tools after machining for 5 min. From the figure, it can be seen that the average temperature of the gel abrasive tool was higher than that of the hot-pressing tool; however, the heat distribution was more uniform in the gel abrasive tool, and the hot-pressing tool had the problem of local overheating, mainly due to its uneven microstructure causing uneven wear on the surface of the tool during grinding. Therefore, the area with little wear applied most of the grinding force, resulting in excessive local grinding heat of the hot-pressing abrasive tool. On the other hand, the wear of the gel abrasive tool was more even, the grinding force on the face of the tool was more evenly distributed, and no local overheating occurred.

Figure 15 shows the sapphire machined by the gel abrasive tool. Figure 16 shows the flatness of the sapphire after processing by the two abrasive tools, which is closely related to the surface shape accuracy of the abrasive tool. As the machining progressed, the flatness of the abrasive tool was copied to the sapphire. It can be seen that the flatness of the sapphire after processing by the gel abrasive tool was slightly more accurate compared to the hot-pressing abrasive tool, mainly because the existence of the binder in the gel abrasive tool had a better ability to hold the abrasive particles, thus minimizing damage, whereas the binding energy of the abrasives in the hot-pressing tool was not strong enough to prevent tool damage, which consequently affected the flatness. Figure 17b shows the microtopography of the sapphire after CMP. It can be seen that there were scratches on the sapphire processed by the hot-pressing abrasive tool, because the agglomerated particles in the tool produced a "plowing" effect on the sapphire during processing. Furthermore, the mechanical effect of the agglomerated particles was too strong, leaving deep scratches on the sapphire surface, which were difficult to remove after CMP. Although there were obvious processing marks left by the gel abrasive tool on the sapphire, they were easily removed after CMP.

**Table 6.** Machining results.

| Group | Spindle Speed (rpm) | Feeding Rate (μm/min) | Grinding Depth (μm) | Surface Roughness $R_a$ (nm) (Average) | |
|---|---|---|---|---|---|
| | | | | Hot-Pressing Tool | Gel Tool |
| 1 | 700 | 8 | 12 | 5.08 | 2.67 |
| 2 | 700 | 8 | 16 | 4.07 | 2.20 |
| 3 | 700 | 10 | 12 | 5.33 | 2.84 |
| 4 | 700 | 10 | 16 | 5.63 | 3.09 |
| 5 | 900 | 8 | 12 | 4.13 | 2.17 |
| 6 | 900 | 8 | 16 | 3.96 | 1.95 |
| 7 | 900 | 10 | 12 | 4.65 | 2.54 |
| 8 | 900 | 10 | 16 | 4.52 | 2.38 |

**Table 7.** Tool wear rate and MRR.

| Group | Tool Wear Rate (μm/h) | | MRR (μm/h) | |
|---|---|---|---|---|
| | Hot-Pressing Tool | Gel Abrasive Tool | Hot-Pressing Tool | Gel Abrasive Tool |
| 1 | 0.76 | 0.19 | 0.57 | 0.43 |
| 2 | 0.84 | 0.24 | 0.62 | 0.48 |
| 3 | 0.95 | 0.25 | 0.63 | 0.51 |
| 4 | 1.07 | 0.28 | 0.66 | 0.54 |
| 5 | 1.12 | 0.31 | 0.63 | 0.56 |
| 6 | 1.36 | 0.36 | 0.71 | 0.59 |
| 7 | 1.72 | 0.39 | 0.77 | 0.61 |
| 8 | 1.94 | 0.46 | 0.76 | 0.66 |

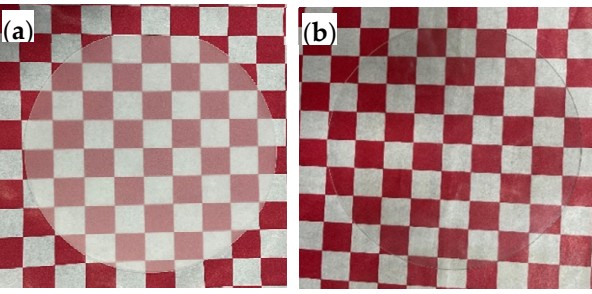

**Figure 15.** Sapphire machined by gel abrasive tool: (**a**) before; (**b**) after.

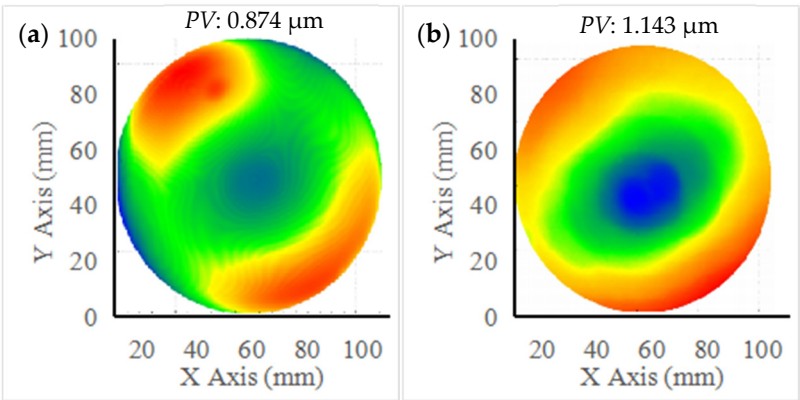

**Figure 16.** Flatness of sapphire after machining using different tools: (**a**) gel abrasive tool; (**b**) hot-pressing abrasive tool.

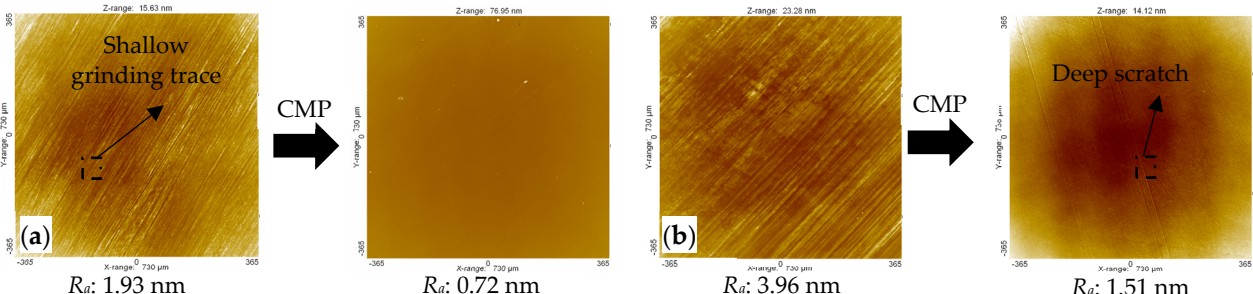

**Figure 17.** Surface topography of the sapphire after machining using different tools and after CMP: (**a**) gel abrasive tool; (**b**) hot-pressing abrasive tool.

*4.5. Comparative Processing Experiments*

The dressing of the abrasive tool greatly influences its performance during the polishing process, affecting not only the surface shape and sharpness of the grinding tools, but also the wear, grinding force, grinding temperature, and surface integrity of the workpiece [31]. The surface morphology of the damaged abrasive tool and the degree of exposure of the abrasive particles change, and these changes are copied to the surface of the workpiece. As a result, the surface quality of the workpiece is affected. To explore the change in surface morphology of the gel abrasive tool during processing, a spindle speed of 900 rpm and feeding rate of 8 µm/min were selected, and grinding was performed in 10 strokes. When the feeding depth reached 16 µm, the damage of the abrasive tool during the grinding process was observed using SEM. It can be seen from Figure 18 that, when the grinding depth of the abrasive tool reached 32 µm, the surface presented a glazed layer. At 64 µm, the phenomenon of glazing became much more severe, the pores of the abrasive tool were blocked by extensive debris due to long-term processing, and the number of exposed abrasive particles on the surface was greatly reduced. When the grinding depth reached 80 µm, the surface of the abrasive tool was glazed. This phenomenon needs to be addressed. The pores on the surface of the abrasive tool were blocked by wear debris, and the blocked pores lost the ability to hold debris and store coolant. Because the frictional heat accumulated during the machining process was difficult to eliminate in time, it likely damaged the abrasive tool. Therefore, when the grinding depth reached 64 µm, it was necessary to dress the abrasive tool. The surface SEM image shows that the exposed abrasive particles on the surface were covered by wear debris. The abrasive tool was trimmed using an electroplated diamond disc. Figure 19 shows the comparison of the abrasive tool before and after modification. It can be seen that the glaze layer on the surface of the abrasive tool disappeared.

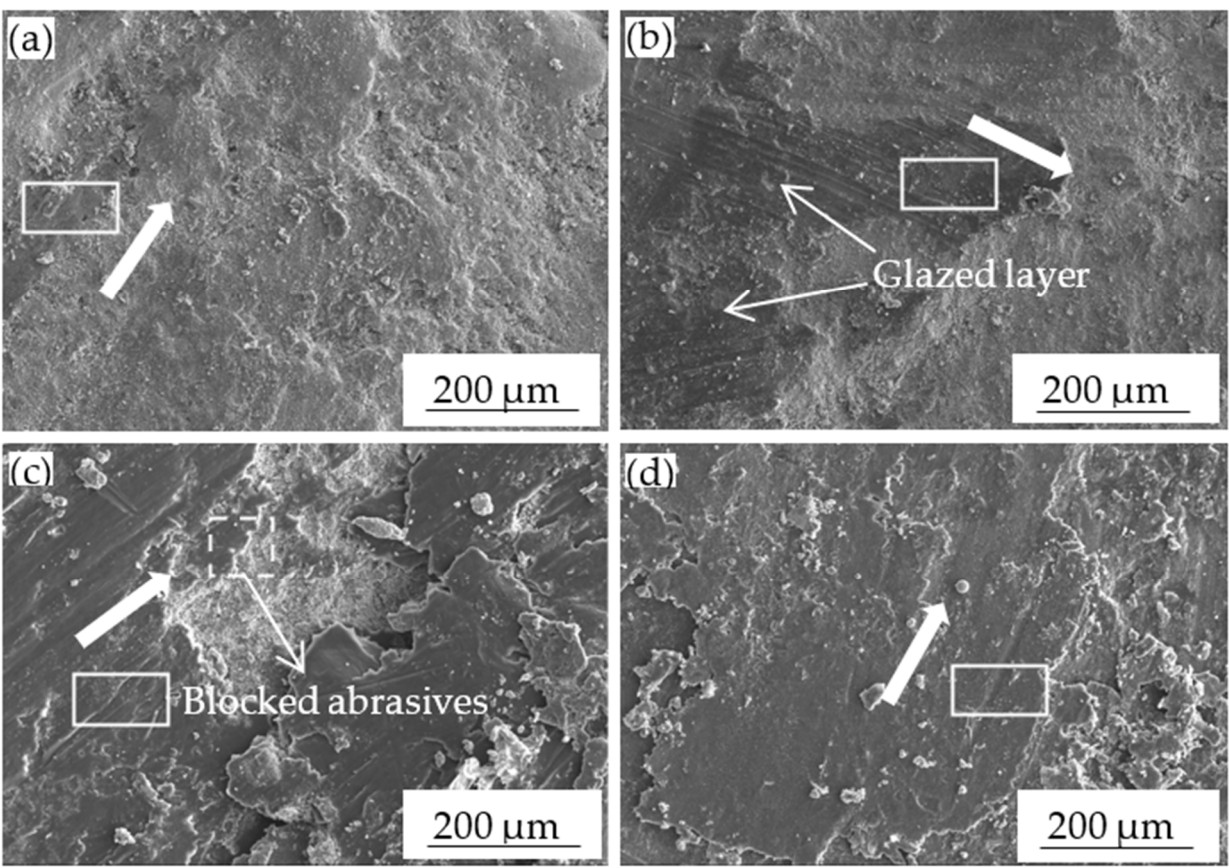

**Figure 18.** Wear state of gel abrasive tool at every stroke: (**a**) 32 μm; (**b**) 48 μm; (**c**) 64 μm; (**d**) 80 μm.

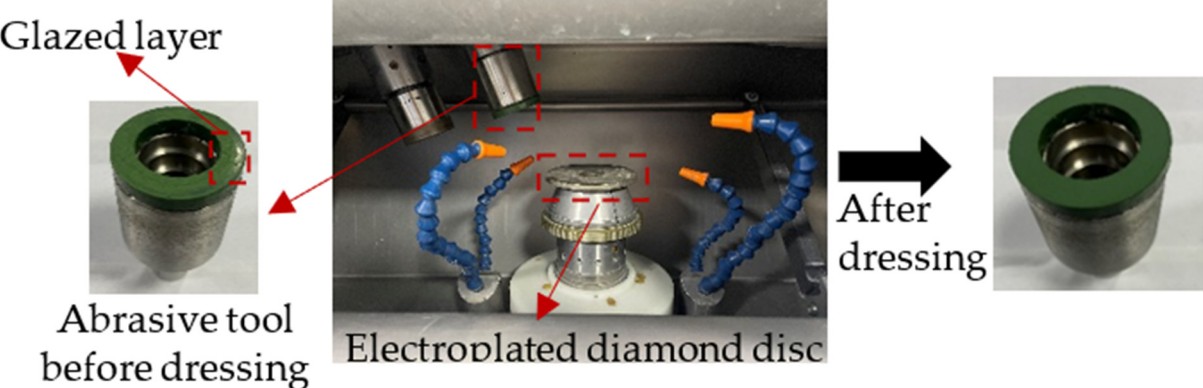

**Figure 19.** Abrasive tool before and after dressing.

## 5. Conclusions

PI resin can reinforce the mechanical properties of a gel abrasive tool; however, when its content is too high, the pre-initiated slurry suffers from poor viscosity. The results from testing the mechanical properties showed that the performance of hot-pressing abrasive tool was worse than that of the gel abrasive tool due to the appearance of loosely distributed agglomerated inorganic particles, which caused the unevenness of the hot-pressing tool. The results of the orthogonal experiment showed that the surface roughness reached 1.88 nm with a spindle speed of 900 rpm, grinding depth of 16 μm, and feeding rate of 8 μm/min. Moreover, the feeding rate had the most significant influence on the surface roughness of the sapphire since it gradually enlarged the contact area between tool and the workpiece, thereby promoting the solid-phase chemical reaction. In the polishing experiment, the

thermogram showed that the hot-pressing tool had a lower average temperature in the machining area; however, the tool suffered from local overheating, whereas the gel abrasive tool had a better processing effect due to its great uniformity. To compare the polishing effect, the CMP method was adopted to manifest the differences. Some deep scratches could be observed from the surface microtopography of the sapphire processed by the hot-pressing tool, whereas the sapphire processed by the gel abrasive tool achieved a scratch-free surface. The tool dressing experiment revealed that, when the total feeding depth reached 64 μm, the abrasive tool suffered from severe glazing, leading to serious blockage. However, after dressing the tool using an electroplated diamond disc, the glazing film disappeared.

**Author Contributions:** T.Z. and K.F. conceptualized and designed the study; T.Z. and K.F. analyzed the experimental data; L.Z. wrote the paper; B.L. and Z.Z. provided guidance and modification of the paper; L.Z. conducted the experiments. All authors have read and agreed to the published version of the manuscript.

**Funding:** The authors gratefully acknowledge the financial support from the Natural Science Foundation of Zhejiang Province (LQ20E050004, LZY22E050007 and LZY21E050004) and the Quzhou Science and Technology Project (No. 2022K88).

**Institutional Review Board Statement:** Not applicable.

**Informed Consent Statement:** Not applicable.

**Data Availability Statement:** The data presented in this study are available on reasonable request from the corresponding author.

**Conflicts of Interest:** The authors declare no conflict of interest.

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
