# Peer review of "Fabrication and Characterization of Gel-Forming Cr2O3 Abrasive Tools for Sapphire Substrate Polishing"

_applsci, doi:10.3390/app122412949_

Round 1

Reviewer 1 Report

The authors present a study on using a gel-formed Cr2O3 abrasive tool to polish sapphire. The authors give sufficient detail of the fabrication method and provide ample results on the influence of polishing process parameters on sapphire surface roughness. Section 4.2, however, suffered from considerable errors making the section unintelligible. A handful of the issues are identified below. Furthermore, how the mechanical properties, namely tensile strength, flexural and impact strength were tested needs addressing. Sample size and surface roughness can have considerable influence on these properties.

·        P. 6, line 181. variables A, B and C do not exist in equation 1. b1, b2 and b3 are the parameters rather indices?

·        P. 6, line 183: Is there a typo in this sentence: “The linear function relationship needs to be deformed to make it a linear function relationship...

·        How are residuals being calculated in figure 4? The explanation given on p. 6, line 188 is not clear.

·        Why does the data look different between the central subplot in figure 5 and figure 6, if presumably they’ve showing the same data (i.e. surface roughness)?

·        Is Figure 4b related to the mechanical properties or surface roughness? The term is not well defined in the caption. Also what is “Case Number” in reference to what? Where is this discussed in the text?

·        Table 5. What is “range analysis”? This needs to be explicitly in the text. No mention of the variables K1, k2, K3, Kalpha1, Kalpha 2 is ever given.

·        P. 6, line 190: is this discussing figure 4b? I don’t see confidence intervals in figure 5. If it is 4b, why is this a subplot of the mechanical data?

Reviewer 2 Report

Abstract:

 please revise opening sentence from line 1-3.

Presentation style of abstract should be improved.

Introduction:

Line 33-42, please remove irrelevant literature and add related to abrasive wear and materials development.

Write down the novelty paragraph at the end of introduction part.

Materials and Methods

PAM + PI resin 15% has been used. What is effect of PAM + PI resin composition variation?

 Results and discussion

Pore structure (Figure 3: line 147-157): cause the production of stress concentrations (mechanical). Therefore, pore density should be calculated and add some data related to micro stress and strain creation effects.

Agglomeration (Figure 3: line 147-157): discuss in detail. Just mentioning is not sufficient.

Figure 4: should include the effect of composition variation on mechanical properties. It is important.

Table 4 and 5 should be discussed in more details. At line 194, the regression constants (b2=0.1580, b3=0.4052) refer to poor behavior. Normally the value should be above 0.8 (at least) for better results.

Figure.6. heat dissipation, temperature effect is missing. That is considered important during cutting.

Figure 7:  Fig 7 (a) and 7 (e) show almost same effect, why? Explain it in article from SEM microstructure point of view.

From line 237-242, authors have provided very general about cause (feed rate) and effect (surface roughness). Better, explain with SEM evaluations. Otherwise, general discussion is not acceptable.

Figure 9:  Fig 9 (a) and 9 (e) show almost same effect, why? Explain it in article from SEM microstructure point of view.  why it appears so same at different feeding rate.

Figure 10 and 11: how much abrasive wear is produced, should present numerically to justify the word “application” in article title. Therefore, please calculate wear and corelate it with figures 10 and 11.

In figure 14a grinding trace and Fig.14 b grinding scratch is not clear.

Strongly recommended to specify direction and wear path in SEM images (Figure 15).

 Conclusions

At the end, conclusion is poorly written.

General Considerations

For general please add at least reasonable number of references in results and discussion section to support your findings and claims. It is strongly recommended.

After reading, your article is only investigations and not supporting any commercial applications. One possible title is as follows:

“Fabrication, mechanical testing, and characterization of Gel-forming Cr2O3 Abrasive 2 Tool”

Round 2

Reviewer 2 Report

Authors have addressed all possbile comments.